# Influence of Integration Schemes and Maneuvers on the Initial Alignment and Calibration of AUVs: Observability and Degree of Observability Analyses

**DOI:** 10.3390/s22093287

**Published:** 2022-04-25

**Authors:** Adriano Frutuoso, Felipe O. Silva, Ettore A. de Barros

**Affiliations:** 1Unmanned Vehicles Laboratory, Department of Mechatronics Engineering, University of Sao Paulo, Avenida Professor Mello Moraes, 2231, Sao Paulo 05508-030, Brazil; eabarros@usp.br; 2Department of Automatics, Federal University of Lavras, Lavras 37200-900, Brazil; felipe.oliveira@ufla.br

**Keywords:** AUV, observability analysis, degree of observability

## Abstract

The use of autonomous underwater vehicles (AUV) has increased in a wide range of sectors, including the oil and gas industry, military, and marine research. The AUV capabilities to operate without a direct human operator and untethered to a support vessel are features that have aroused interest in the marine environment. The localization of AUV is significantly affected by the initial alignment and the calibration of the navigation sensors. In this sense, this paper proposes a thorough observability analysis applied to the latter problem. The observability analysis is carried out considering three types of sensor fusion integration and a set of maneuvers, and the results are validated through numerical simulations. As main contribution of this paper, it is shown how the addition of position errors in the observation vector can decouple some gyro and accelerometer biases from the latitude and altitude errors, particularly in the stationary observability analysis. The influence of oscillations in the diving plane and typical AUV maneuvers are analyzed, showing their relative impacts on the degree of observability of the inertial measurement unit (IMU)/Doppler velocity log (DVL) misalignment and DVL scale factor error. Finally, the state’s estimation accuracy is also analyzed, showing the limitation of the degree of observability as an assessment tool for the estimability of the states.

## 1. Introduction

Technological advances in recent years have promoted the development of autonomous vehicles for a wide variety of sectors, including aerial, aquatic, and terrestrial applications. Internet of Things and smart cities [1,2] inspection of oil and gas pipelines [3], swarm of underwater robots [4], and autonomous cars [5] are some examples of recent applications of autonomous vehicles.

Unlike terrestrial and aerial mobile robots, autonomous underwater vehicles (AUV) should deal with particular boundaries related to the limited transmission of electromagnetic signals underwater. Those limits consist of the absence of the localization measures provided by satellite-based systems, and the absence of a high band and long range communication with the human user who is at the surface. The first kind of limitation has an impact at AUV self-localization, which is the subject of this work.

The initial alignment, which consists of determining the initial orientation of the vehicle, and the calibration of the navigation sensors are key factors that significantly influence the performance of AUV navigation.

The inertial measurement unit (IMU), Doppler velocity log (DVL), pressure sensor (PS), and global positioning system (GPS) are navigation sensors widely used in AUVs [6]. Therefore, the proper estimation of the navigation sensor errors, e.g., inertial sensors biases, DVL scale factor error, and the misalignment between the IMU and the DVL, as well as the alignment errors, are necessary to improve navigation performance.

The dynamic and observation models and the movements executed by the AUV affect the performance of the estimation. In this sense, the observability analysis is important to determine the efficiency of the Kalman filter in the estimation of the states [7].

A system is said to be observable if the initial states, at time $t_{0}$, can be determined at time $t_{1}$, from observations of the inputs and outputs of the system taken at [$t_{0}$, $t_{1}$] [8]. The observability analysis can be classified as qualitative and quantitative [9]. The qualitative analysis is a binary criterion, defining only whether the system or its states are observable or unobservable. This criterion was first proposed by Kalman [10]. The quantitative analysis, in turn, defines how observable the system or its states are. This analysis is related to the concept of the degree of observability.

According to Friedman [9], the rank of the observability matrix, its linearly independent columns, eigenvalues, singular values, and determinant are some metrics used to analyze the observability of a system. In particular, the matrix rank is extensively employed in the qualitative analysis. Concerning the degree of observability, it can be calculated using the degree of independence between the columns (or the singular values) of the observability matrix/Grammian. The singular values, in particular, are largely employed to determine the degree of observability numerically [11,12,13,14,15].

In the last years, observability analysis has been employed to the AUV navigation problem, especially during its initial alignment and calibration phase, to evaluate the observability of the navigation variables subject to maneuver conditions [14,16,17,18,19,20]. In [16], it was shown that the observability of the DVL velocity error in the navigation frame is influenced by the AUV angular movement. However, the dynamic model used in the analysis did not include the vertical velocity error and the altitude error. Furthermore, the misalignment between the IMU and the DVL and the DVL scale factor error were not directly included in the dynamic model.

In [17], analytical expressions were derived for the UnObservable Subspace (UOS) of the AUV navigation problem, considering the following maneuvers: (1) stationary, (2) stationary with yaw angle variation, and (3) constant speed with yaw angle variation. The UOS derivation was performed based on the determination of the kernel of the observability Grammian. However, the IMU/DVL misalignment and the DVL scale factor error were, again, not included in the analysis. As a consequence, in the inertial navigation system (INS)/DVL integration for the AUV static condition, the dimension of the UOS was found to be 4, comprising the gyro bias error in the z-axis, the vertical alignment error, and linear combinations of the north and east alignment errors with the accelerometer bias errors in the x and y axes. Such a UOS dimension, however, is in disagreement with other works [21,22], for which the dimension of the UOS, in the stationary case, was 3 (obviously disregarding the position errors). For Bar-Itzhack and Berman [21], and Silva, Hemerly, and Leite Filho [22], such UOS comprises linear combinations of the accelerometer bias errors in the north and east directions with the alignment errors in the north and east, plus a linear combination of the vertical alignment error with the gyro bias error in the east direction.

In addition, the dynamic model of INS error propagation presented in [17] did not include the coupling term between the attitude error vector and the velocity error vector, which contributed to the attainment of a UOS dimension equal to 4. In this sense, the dynamic model of [17] can be seen as a simplification of the dynamic model presented in [23], particularly chosen to facilitate the computation of the kernel of the observability Grammian.

In [18], it was demonstrated that trajectories containing segments of type I (constant attitude with linearly independent specific force variation in the body frame) and type II (turning segments) make the attitude errors, the velocity errors, the inertial sensors biases, the IMU/DVL misalignment, and the DVL scale factor error observable. However, the degrees of observability of the state variables were not evaluated. The results of [19] pointed out that the addition of vertical and translational movements in the AUV trajectory make the variables associated with the DVL calibration completely observable. The results presented in [14,20] further show that the degrees of observability of the state variables increased with changes in the direction of the trajectory.

In [14,19,20], however, the authors did not address the couplings between the state variables, which are not distinguished by the Kalman filter [21]. Furthermore, in [19], the variables associated with the initial alignment and the biases of the inertial sensors were not included in the observability analysis. In [20], in turn, the IMU/DVL misalignment was disregarded from the analysis, while in [14], the degrees of observability of the DVL scale factor errors and the IMU/DVL misalignment were not evaluated.

Given the aforementioned open issues, this work performs quantitative and qualitative observability analyses regarding the problem of the initial alignment and the calibration of the navigation sensors of the AUV. The quantitative analysis is performed by the rank computation, and the qualitative analysis is performed by the degree of observability computation. The observability analysis is carried out considering three types of sensor fusion integration and a set of maneuvers: (1) stationary, (2) mooring, (3) straight-line with constant speed, (4) straight-line with acceleration, (5) zigzag, (6) lawn mower with constant speed, and (7) lawn mower with zigzag and swing. The observability analysis results are particularly useful in the AUV mission planning, in terms of the choice of the better maneuver and the definition of the appropriated integration.

Numerical simulations are carried out to generate the results. In the stationary observability analysis, it is shown how the addition of position errors in the observation vector can decouple some gyro and accelerometer biases from the latitude and altitude errors. The influence of oscillations in the diving plane and typical AUV maneuvers is also analyzed, showing their relative impacts on the degree of observability of the IMU/DVL misalignment and DVL scale factor error. In addition, the accuracy of the state estimation is also analyzed, showing the limitation of the degree of observability as an assessment tool for the estimability of the states.

This paper is organized as follows: Section 2 presents the dynamic model and the observation models for the INS/GPS/DVL/PS, INS/DVL/PS, and INS/DVL integrations. In Section 3, the observability decomposition is presented. Section 4 is dedicated to the observability analysis of the AUV linear time-varying system using the stripped observability matrix and the singular value decomposition for the rank computation. Section 5 is dedicated to the evaluation of the corresponding degree of observability. Section 6 and Section 7, lastly, present simulation results and the discussions, respectively. Section 8 is dedicated to the conclusions.

## 2. System Modelling

The expanded INS errors resolved in the NED (north-east-down) frame is given by Equation (1) [24,25], where $x={\left[\begin{array}{ccccccccc} \phi^{n} & \delta \upsilon^{n} & \delta L & \delta \lambda & \delta h & b_{g} & b_{a} & e & s_{f} \end{array}\right]}^{T}$ is the state vector; $\phi^{n}={\left[\begin{array}{ccc} \phi_{N} & \phi_{E} & \phi_{D} \end{array}\right]}^{T}$ corresponds to the attitude error vector; $\delta \upsilon^{n}$ is the velocity error vector; δL, δλ, δh are the latitude, longitude e altitude errors, respectively; $b_{g}$ and $b_{a}$ are the gyro and accelerometer bias; $w_{g}$ and $w_{a}$ are the gyro and accelerometer noises, respectively; $e={\left[\begin{array}{ccc} e_{x} & e_{y} & e_{z} \end{array}\right]}^{T}$ and $s_{f}$ correspond to the IMU/DVL misalignment errors and scale factor error of the DVL, respectively; $C_{b}^{n}$ is the attitude matrix from the body frame to the navigation frame; $\omega_{in}^{n}$ is the angular rate of the navigation frame with respect to the inertial frame resolved in the navigation frame; and $f^{n}$ is the specific force resolved in the navigation frame.

The coefficients of the INS error model are given by Equations (2)–(13), where $R_{N}$ and $R_{E}$ are the meridian and transverse radii of curvature, respectively; g is the local gravity; and Ω is the turn rate of the Earth. $R_{0}$ is computed as $R_{0}=\sqrt{R_{N}R_{E}}$.
(1)$$\dot{x}=[\begin{array}{cccccccc} -[\omega_{in}^{n}\times ] & A_{1} & A_{2} & 0_{3\times 1} & A_{3} & -C_{b}^{n} & 0_{3\times 3} & 0_{3\times 4}\\ \left[f^{n}\times \right] & \left([\upsilon^{n}\times ]A_{1}-A_{4}\right) & [\upsilon^{n}\times ]A_{5} & 0_{3\times 1} & \left([\upsilon^{n}\times ]A_{3}+C_{5}\right) & 0_{3\times 3} & C_{b}^{n} & 0_{3\times 4}\\ 0_{1\times 3} & B_{1} & 0 & 0 & b_{2} & 0_{1\times 3} & 0_{1\times 3} & 0_{1\times 4}\\ 0_{1\times 3} & C_{1} & c_{2} & 0 & c_{3} & 0_{1\times 3} & 0_{1\times 3} & 0_{1\times 4}\\ 0_{1\times 3} & C_{4} & 0 & 0 & 0 & 0_{1\times 3} & 0_{1\times 3} & 0_{1\times 4}\\ 0_{3\times 3} & 0_{3\times 3} & 0_{3\times 1} & 0_{3\times 1} & 0_{3\times 1} & 0_{3\times 3} & 0_{3\times 3} & 0_{3\times 4}\\ 0_{3\times 3} & 0_{3\times 3} & 0_{3\times 1} & 0_{3\times 1} & 0_{3\times 1} & 0_{3\times 3} & 0_{3\times 3} & 0_{3\times 4}\\ 0_{4\times 3} & 0_{4\times 3} & 0_{4\times 1} & 0_{4\times 1} & 0_{4\times 3} & 0_{4\times 3} & 0_{4\times 3} & 0_{4\times 4} \end{array}]x+[\begin{array}{cc} -C_{b}^{n} & 0_{3\times 3}\\ 0_{3\times 3} & C_{b}^{n}\\ 0_{1\times 3} & 0_{1\times 3}\\ 0_{1\times 3} & 0_{1\times 3}\\ 0_{1\times 3} & 0_{1\times 3}\\ 0_{3\times 3} & 0_{3\times 3}\\ 0_{3\times 3} & 0_{3\times 3}\\ 0_{4\times 3} & 0_{4\times 3} \end{array}][\begin{array}{c} w_{g}\\ w_{a} \end{array}]$$
(2)$$A_{1}=\left[\begin{array}{ccc} 0 & \frac{1}{R_{E}+h} & 0\\ -\frac{1}{R_{N}+h} & 0 & 0\\ 0 & -\frac{\tan\left(L\right)}{R_{E}+h}\; & 0 \end{array}\right]$$
(3)$$A_{2}=\left[\begin{array}{c} -\Omega \;\sin\left(L\right)\\ 0\\ -\Omega \cos\left(L\right)-\frac{\upsilon_{E}\sec^{2}\left(L\right)}{R_{E}+h} \end{array}\right]$$
(4)$$A_{3}=\left[\begin{array}{c} -\frac{\upsilon_{E}}{{\left(R_{E}+h\right)}^{2}}\\ \frac{\upsilon_{N}}{{\left(R_{N}+h\right)}^{2}}\\ \frac{\upsilon_{E}\tan\left(L\right)}{{\left(R_{E}+h\right)}^{2}} \end{array}\right]$$
(5)$$A_{4}=\left[\begin{array}{ccc} 0 & 2\Omega \sin\left(L\right)+\frac{\upsilon_{E}\tan\left(L\right)}{R_{E}+h} & -\frac{\upsilon_{N}}{R_{N}+h}\\ -2\Omega \sin\left(L\right)-\frac{\upsilon_{E}\tan\left(L\right)}{R_{E}+h} & 0 & -2\Omega \cos\left(L\right)-\frac{\upsilon_{E}}{R_{E}+h}\\ \frac{\upsilon_{N}}{R_{N}+h} & 2\Omega \cos\left(L\right)+\frac{\upsilon_{E}}{R_{E}+h} & 0 \end{array}\right]$$
(6)$$A_{5}=\left[\begin{array}{c} -2\Omega \sin\left(L\right)\\ 0\\ -2\Omega \cos\left(L\right)-\frac{\upsilon_{E}\sec^{2}\left(L\right)}{R_{E}+h} \end{array}\right]$$
(7)$$B_{1}=\left[\begin{array}{ccc} \frac{1}{R_{N}+h} & 0 & 0 \end{array}\right]$$
(8)$$b_{2}=-\frac{\upsilon_{N}}{{\left(R_{N}+h\right)}^{2}}$$
(9)$$C_{1}=\left[\begin{array}{ccc} 0 & \frac{\sec\left(L\right)}{R_{E}+h} & 0 \end{array}\right]$$
(10)$$c_{2}=\frac{\sec\left(L\right)\tan\left(L\right)\upsilon_{E}}{R_{E}+h}$$
(11)$$c_{3}=-\frac{\upsilon_{E}\sec\left(L\right)}{{\left(R_{E}+h\right)}^{2}}$$
(12)$$C_{4}=\left[\begin{array}{ccc} 0 & 0 & -1 \end{array}\right]$$
(13)$$C_{5}=\left[\begin{array}{c} 0\\ 0\\ -\frac{2g}{R_{0}+h} \end{array}\right]$$

Considering the INS/GPS/DVL/PS loosely coupled integrated navigation, the observation model is as follows: (14)$$y=\left[\begin{array}{c} \delta \upsilon^{n}\\ \delta L\\ \delta \lambda \\ \delta h \end{array}\right]=\left[\begin{array}{c} \upsilon^{n}-{\tilde{\upsilon }}_{d}^{n}\\ L-{\tilde{L}}_{gps}\\ \lambda -{\tilde{\lambda }}_{gps}\\ h-{\tilde{h}}_{ps} \end{array}\right]$$
(15)$$y=\mathrm{Hx}+\nu =\left[\begin{array}{ccccccc} -\left[\upsilon^{n}\times \right] & I_{3\times 3} & 0_{3\times 3} & 0_{3\times 3} & 0_{3\times 3} & -C_{b}^{n}\left[\upsilon^{b}\times \right] & -\upsilon^{n}\\ 0_{3\times 3} & 0_{3\times 3} & I_{3\times 3} & 0_{3\times 3} & 0_{3\times 3} & 0_{3\times 3} & 0_{3\times 1} \end{array}\right]x+\nu$$
where ${\tilde{\upsilon }}_{d}^{n}$ represents the DVL velocity vector resolved in the navigation frame; ${\tilde{L}}_{gps}$ and ${\tilde{\lambda }}_{gps}$ are the GPS latitude and longitude, respectively; ${\tilde{h}}_{ps}$ is the altitude from the PS measurements; and ν is the measurement noise vector.

Considering the loosely coupled INS/DVL/PS integration, the vector and the observation model are as follows:(16)$$y=\left[\begin{array}{c} \delta \upsilon^{n}\\ \delta h \end{array}\right]=\left[\begin{array}{c} \upsilon^{n}-{\tilde{\upsilon }}_{d}^{n}\\ h-{\tilde{h}}_{ps} \end{array}\right]$$
(17)$$y=\mathrm{Hx}+\nu =\left[\begin{array}{ccccccc} -\left[\upsilon^{n}\times \right] & I_{3\times 3} & 0_{3\times 3} & 0_{3\times 3} & 0_{3\times 3} & -C_{b}^{n}\left[\upsilon^{b}\times \right] & -\upsilon^{n}\\ 0_{1\times 3} & 0_{1\times 3} & \left[\begin{array}{ccc} 0 & 0 & 1 \end{array}\right] & 0_{1\times 3} & 0_{1\times 3} & 0_{1\times 3} & 0 \end{array}\right]x+\nu$$

For the loosely coupled INS/DVL integration:(18)$$y=\delta \upsilon^{n}=\upsilon^{n}-{\tilde{\upsilon }}_{d}^{n}$$
(19)$$y=\mathrm{Hx}+\nu =\left[\begin{array}{ccccccc} -\left[\upsilon^{n}\times \right] & I_{3\times 3} & 0_{3\times 3} & 0_{3\times 3} & 0_{3\times 3} & -C_{b}^{n}\left[\upsilon^{b}\times \right] & -\upsilon^{n} \end{array}\right]x+\nu$$

## 3. Observability Decomposition 

Consider the linear and time-invariant (LTI) model:(20)$$\left\{\begin{array}{c} \;\dot{x}\;\left(t\right)=Ax\left(t\right)+Bu\left(t\right)\\ y\left(t\right)=Hx\left(t\right)+Du\left(t\right) \end{array}\right.$$

If the system is unobservable, there is a similarity transformation, shown in Equation (21), which results in the decomposed form of Equation (22), where the subsystem $\left(A_{11},B_{1},H_{1},D\right)$ is observable [8]. $A_{11}$ is a square matrix r×r, where r is the dimension of the observable subspace.
(21)$$\left\{\begin{array}{c} \;\bar{x}=Tx\\ \;\bar{A}=TAT^{-1}\\ \;\bar{B}=TB\\ \;\bar{H}=HT^{-1}\\ \;\bar{D}=D \end{array}\right.\;$$
(22)$$\left\{\begin{array}{c} \overset{\dot{¯}}{x}\left(t\right)=\left[\begin{array}{cc} A_{11} & 0\\ A_{21} & A_{22} \end{array}\right]\;\bar{x}\left(t\right)+\left[\begin{array}{c} B_{1}\\ B_{2} \end{array}\right]u\left(t\right)\\ y\left(t\right)=\left[\begin{array}{cc} H_{1} & 0 \end{array}\right]\;\bar{x}\left(t\right)+Du\left(t\right) \end{array}\right.$$

The new state vector $\bar{x}$ corresponds to combinations of the observable and unobservable states of the original system. The similarity transformation does not change the observability of the original system [8].

The similarity transformation matrix is given by Equation (23), where $\langle v_{1},{\;v}_{2},\;...,{\;v}_{r}\rangle$ is a set of linearly independent vectors of ${ℜ}^{n}$. $\langle v_{r+1},{\;v}_{r+2},\;...,{\;v}_{n}\rangle$ is a base of the kernel of the observability matrix $Q_{o}$:(23)$$T={\left[\begin{array}{ccccccc} v_{1} & v_{2} & \ldots & v_{r} & v_{r+1} & \ldots & v_{n} \end{array}\right]}^{-1}$$
(24)$$Q_{o}v_{i}=0,i=r+1,\ldots ,n$$
where
(25)$$Q_{o}=\left[\begin{array}{c} H\\ \mathrm{HA}\\ {\mathrm{HA}}^{2}\\ \vdots \\ {\mathrm{HA}}^{n-1} \end{array}\right]$$

## 4. Stripped Observability Matrix and Singular Value Decomposition 

Consider the linear time-varying (LTV) model for the continuous and discrete cases:(26)$$\left\{\begin{array}{c} \dot{x}\left(t\right)=A\left(t\right)x\left(t\right)\\ y\left(t\right)=H\left(t\right)x\left(t\right) \end{array}\right.$$
(27)$$\left\{\begin{array}{c} x_{k+1}=\Phi_{k}x_{k}\\ y_{k}=H_{k}x_{k} \end{array}\right.$$

The observability of an LTV system in the time interval $t_{o}$ to $t_{f}$ can be determined from the analysis of the rank of the observability Grammian matrix O. The observability Grammian matrices for the continuous and discrete cases are as follows [9]:(28)$$O\left(t_{o},t_{f}\right)=\overset{t_{f}}{\underset{t_{o}}{\int }}\Phi {\left(t,t_{o}\right)}^{T}H{\left(t\right)}^{T}H\left(t\right)\Phi \left(t,t_{o}\right)dt$$
(29)$$O\left(k_{o},k_{f}\right)=\overset{k_{f}}{\underset{k=k_{0}}{\sum }}\Phi {\left(k,k_{f}\right)}^{T}H_{k}^{T}H_{k}\Phi \left(k,k_{f}\right)$$

Equations (28) and (29) show that for the determination of O, it is necessary to calculate the state transition matrices, which are $\Phi \left(t,t_{o}\right)$ and $\Phi \left(k,k_{f}\right)$. If O is a non-singular matrix, the LTV system is observable in the considered time interval.

In general, the calculation of O is performed numerically [7,9]. As an alternative to O, the observability of the LTV system can be evaluated by the rank of $Q_{\mathrm{SOM}}$ (stripped observability matrix), which is obtained by partitioning the LTV system into piece-wise constant systems (PWCS) [7].

Then, consider the PWCS models for the discrete case:(30)$$\left\{\begin{array}{c} x_{k+1}=\Phi_{j}x_{k}\\ y_{j}=H_{j}x_{k} \end{array}\right.$$

For each j time segment, matrices $\Phi_{j}$ and $H_{j}$ are constant. Then, $Q_{\mathrm{SOM}}$ is computed using Equation (31), where $Q_{o,j}$, for j=1, 2,…, l, corresponds to the observability matrix for each j time segment.
(31)$$Q_{\mathrm{SOM}}\left(r\right)=\left[\begin{array}{c} Q_{o,1}\\ Q_{o,2}\;\\ \vdots \\ Q_{o,l} \end{array}\right]$$

The rank of a matrix can be numerically determined by singular value decomposition into (SVD) or by decomposition into eigenvalues. The rank corresponds to the number of singular values or eigenvalues different from zero [9]. Unlike eigenvalues, singular values are always equal to or greater than zero.

The SVD of a matrix $A\in {ℜ}^{m\times n}$ is [13,15]:(32)$$A={U\Sigma V}^{T}$$
where $U=\left[\begin{array}{cccc} u_{1} & u_{2} & \ldots & u_{m} \end{array}\right]$ and $V=\left[\begin{array}{cccc} v_{1} & v_{2} & \ldots & v_{n} \end{array}\right]$ are the orthogonal matrices, $\Sigma =diag\left(S,0\right)$, $\Sigma \in R^{m\times n}$, is a matrix of singular values, and $S=diag\left(\sigma_{1},\sigma_{2},\sigma_{3},\ldots ,\sigma_{n}\right)$, with $\sigma_{1}\geq \sigma_{2}\geq \sigma_{3}\geq \ldots \geq \sigma_{n}$, which corresponds to a diagonal matrix containing the singular values of A.

Then, an LTV system is observable if $Q_{\mathrm{SOM}}$ is full rank. On the contrary, if the SVD of $Q_{\mathrm{SOM}}$ results in null singular values, the system is unobservable.

## 5. Degree of Observability 

The rank of $Q_{\mathrm{SOM}}$ only informs whether the system is observable or not. The observability condition of the states, in particular, cannot be evaluated by the rank analysis. For LTI systems, the observability of the states can be obtained directly from analytical expressions, using similarity transformations that decompose the system into observable and unobservable subspaces [8,21]. However, the observation model and the dynamic model of the navigation errors, presented in Section 2, are LTV. So, it is hard to obtain analytical expressions for the observability of the states.

Alternatively, the concept degree of observability can be used. The degree of observability informs how observable the state of the system is.

Usual methods to evaluate the observability degree of a state rely on the $Q_{\mathrm{SOM}}$ singular value decomposition approach [11,13,15,26]. In particular, the degree of observability can be determined using the weighted singular vector $u_{r}$ [11,26]:(33)$$u_{r}=\overset{r}{\underset{i=1}{\sum }}\frac{\sigma_{i}}{\sigma_{1}}u_{i}$$

Equation (33) represents the weighted singular vector for the first r singular vectors of U. The elements of a greater magnitude of $u_{r}$ correspond to the states with a higher degree of observability, which, hence, can be determined as follows:(34)$$n_{j}=\frac{u_{r}\left(j\right)}{u_{r0}}$$
where $u_{r0}$ corresponds to the elements of $u_{r}$ associated with the measured states.

Note that the correspondence of the elements of $u_{r}$ with the degree of observability is direct and independent of the measurements. In practical terms, the weighted singular vector allows for computing the degree of observability using only the observability matrix of the system.

## 6. Results

This section presents simulation results aiming at the observability analysis of the initial alignment and calibration of the AUV navigation sensors, subject to the following maneuver conditions:
Stationary.Mooring. For a mooring maneuver, the vehicle motion was modeled with oscillations in roll (ϕ), pitch (θ), and yaw (ψ), with the same amplitude and period, and with oscillatory translational movement in the surge (x), sway (y), and heave (z) axes, also with the same amplitude and period. For roll, pitch, and yaw, the model is given by the following (in deg):$$\phi ,\theta ,\psi =0^{\circ }+5.0^{\circ }\sin\left(\frac{2\pi t}{10}\right)$$For the translational movements (in m/s):$$\upsilon_{x},\upsilon_{y},\upsilon_{z}=0+0.1\sin\left(\frac{2\pi t}{10}\right)$$Straight-line with constant speed. In the straight-line maneuver with a constant speed, the velocity was set to 1.0 m/s in the x-axis, and 0 in the y and z axes. The swing movement was disregarded.Straight-line with acceleration. In this maneuver, the velocities in the y and z axes, as well as the oscillations in roll, pitch, and yaw, were null. The velocity model in the x-axis is given by the following (in m/s):$$\upsilon_{x}=1.5+0.75\sin\left(\frac{2\pi t}{10}\right)$$Therefore, the resulting acceleration in the x-axis (in m/s^2^) is as follows: $a_{x}=\frac{1.5\pi }{10}\cos\left(\frac{2\pi t}{10}\right)$.Zigzag. In the zigzag maneuver, the speed was set to 1.0 m/s in the x-axis, and 0 in the y and z axes. The roll and pitch oscillations were 0. The yaw oscillation model is given by the following (in deg): $\psi =0^{\circ }+5.0^{\circ }\sin\left(\frac{2\pi t}{10}\right)$.Lawn mower with constant speed. In the lawn mower maneuver, the velocity was set to 1.0 m/s in the x-axis, and zero in the y and z axes. The swing movement was disregarded.Lawn mower with zigzag and swing. For this maneuver, roll and pitch oscillations were null. For yaw, the oscillation model is given by the following (in deg):$$\psi =0^{\circ }+5.0^{\circ }\sin\left(\frac{2\pi t}{10}\right)$$

The velocity models (in m/s) are given by the following:$$\upsilon_{x}=1.0+0.05\sin\left(\frac{2\pi t}{10}\right)$$
$$\upsilon_{y}=\upsilon_{z}=0.1\sin\left(\frac{2\pi t}{10}\right)$$

All maneuvers lasted 1 h. For the stationary maneuver, analytical expressions were obtained for the observability of the states, as, for this condition, the dynamic and observation models are LIT. Regarding the degrees of observability of the state variables, they were calculated using Equation (34). The following assumptions were additionally considered:AUV aligned with the navigation frame (NED).Initial coordinates of −23 degrees (latitude) and −45 degrees (longitude).Zero initial altitude.IMU/DVL misalignment and DVL scale factor error are null.Loosely coupled INS/GPS/DVL/PS, INS/DVL/PS, and INS/DVL integrations, with dynamic and observational models, as given by Equations (1), (15), (17), and (19).

### 6.1. Observability Decomposition: Stationary Case

For the INS/GPS/DVL/PS integration, the rank of $Q_{o}$ is 12. So, the dimension of the unobservable subspace is equal to 7. Applying the similarity transformation given by Equation (35), where W is a matrix of r vectors of the canonical base of ${ℜ}^{19}$ and linearly independent of the kernel of $Q_{o}$.
(35)$$\bar{x}=\mathrm{Tx}=\left[\begin{array}{c} {\bar{x}}_{o}\\ {\bar{x}}_{\bar{o}} \end{array}\right]$$
(36)$$T={\left[\begin{array}{cc} W & \ker\left(Q_{o}\right) \end{array}\right]}^{-1}$$
(37)$$\ker(Q_{0})=[\begin{array}{ccccccc} 0 & 0 & \frac{1}{g} & 0 & 0 & 0 & 0\\ 0 & -\frac{1}{g} & 0 & 0 & 0 & 0 & 0\\ \frac{1}{\Omega \cos(L)} & 0 & -\frac{\tan(L)}{g} & 0 & 0 & 0 & 0\\ 0 & 0 & 0 & 0 & 0 & 0 & 0\\ 0 & 0 & 0 & 0 & 0 & 0 & 0\\ 0 & 0 & 0 & 0 & 0 & 0 & 0\\ 0 & 0 & 0 & 0 & 0 & 0 & 0\\ 0 & 0 & 0 & 0 & 0 & 0 & 0\\ 0 & 0 & 0 & 0 & 0 & 0 & 0\\ 0 & \frac{\Omega \sin(L)}{g} & 0 & 0 & 0 & 0 & 0\\ 1 & 0 & 0 & 0 & 0 & 0 & 0\\ 0 & \frac{\Omega \cos(L)}{g} & 0 & 0 & 0 & 0 & 0\\ 0 & 1 & 0 & 0 & 0 & 0 & 0\\ 0 & 0 & 1 & 0 & 0 & 0 & 0\\ 0 & 0 & 0 & 0 & 0 & 0 & 0\\ 0 & 0 & 0 & 1 & 0 & 0 & 0\\ 0 & 0 & 0 & 0 & 1 & 0 & 0\\ 0 & 0 & 0 & 0 & 0 & 1 & 0\\ 0 & 0 & 0 & 0 & 0 & 0 & 1 \end{array}]$$

The following states and combinations of states are found to be observable:
(38)$${\bar{x}}_{o}=[\begin{array}{c} \phi_{N}-\frac{b_{ay}}{g}\\ \phi_{E}+\frac{b_{ax}}{g}\\ \phi_{D}-\frac{b_{gy}}{\Omega \cos(L)}+\frac{\tan(L)b_{ay}}{g}\\ \delta \upsilon_{N}\\ \delta \upsilon_{E}\\ \delta \upsilon_{D}\\ \delta L\\ \delta \lambda \\ \delta h\\ b_{gx}-\frac{\Omega \sin(L)b_{ax}}{g}\\ b_{gz}-\frac{\Omega \cos(L)b_{ax}}{g}\\ b_{az} \end{array}]$$

The unobservable states, in turn, are as follows:
(39)$${\bar{x}}_{o}=[\begin{array}{c} b_{gy}\\ b_{ax}\\ b_{ay}\\ e_{x}\\ e_{y}\\ e_{z}\\ s_{f} \end{array}]$$

As can be inferred, all of the states are individually unobservable for the stationary condition, except for the velocity errors, the position errors, and the accelerometer bias in the z-axis ($b_{az}$).

Considering the INS/DVL/PS integration, the rank of $Q_{o}$ is equal to 10, hence the dimension of the unobservable subspace becomes 9. Applying the similarity transformation, the kernel of $Q_{o}$ is:
(40)$$\ker(Q_{0})=[\begin{array}{ccccccccc} 0 & 0 & 0 & 0 & \frac{1}{g} & 0 & 0 & 0 & 0\\ 0 & 0 & 0 & -\frac{1}{g} & 0 & 0 & 0 & 0 & 0\\ 0 & \frac{1}{\Omega \cos(L)} & 0 & 0 & -\frac{tan(L)}{g} & 0 & 0 & 0 & 0\\ 0 & 0 & 0 & 0 & 0 & 0 & 0 & 0 & 0\\ 0 & 0 & 0 & 0 & 0 & 0 & 0 & 0 & 0\\ 0 & 0 & 0 & 0 & 0 & 0 & 0 & 0 & 0\\ 0 & 0 & -\frac{1}{\Omega \cos(L)} & \frac{1}{g} & 0 & 0 & 0 & 0 & 0\\ 1 & 0 & 0 & 0 & 0 & 0 & 0 & 0 & 0\\ 0 & 0 & 0 & 0 & 0 & 0 & 0 & 0 & 0\\ 0 & 0 & tan(L) & 0 & 0 & 0 & 0 & 0 & 0\\ 0 & 1 & 0 & 0 & 0 & 0 & 0 & 0 & 0\\ 0 & 0 & 1 & 0 & 0 & 0 & 0 & 0 & 0\\ 0 & 0 & 0 & 1 & 0 & 0 & 0 & 0 & 0\\ 0 & 0 & 0 & 0 & 1 & 0 & 0 & 0 & 0\\ 0 & 0 & 0 & 0 & 0 & 0 & 0 & 0 & 0\\ 0 & 0 & 0 & 0 & 0 & 1 & 0 & 0 & 0\\ 0 & 0 & 0 & 0 & 0 & 0 & 1 & 0 & 0\\ 0 & 0 & 0 & 0 & 0 & 0 & 0 & 1 & 0\\ 0 & 0 & 0 & 0 & 0 & 0 & 0 & 0 & 1 \end{array}]$$

The observable states and combinations of states are, then:
(41)$${\bar{x}}_{o}=[\begin{array}{c} \phi_{N}-\frac{b_{ay}}{g}\\ \phi_{E}-\frac{b_{ax}}{g}\\ \phi_{D}-\frac{b_{gy}}{\Omega \cos(L)}+\frac{\tan(L)b_{ay}}{g}\\ \delta \upsilon_{N}\\ \delta \upsilon_{E}\\ \delta \upsilon_{D}\\ \delta_{h}\\ b_{gx}+\Omega \sin(L)\delta L-\frac{\Omega \sin(L)b_{ax}}{g}\\ b_{gx}+\Omega \cos(L)\delta L-\frac{\Omega \cos(L)b_{ax}}{g}\\ b_{az} \end{array}]$$

The unobservable states and combinations of states, in turn, are as follows:
(42)$${\bar{x}}_{o}=[\begin{array}{c} \delta \lambda \\ b_{gy}\\ \frac{\Omega cos(L)b_{ax}}{g}-\Omega cos(L)\delta L\\ b_{ax}\\ b_{ay}\\ e_{x}\\ e_{y}\\ e_{z}\\ s_{f} \end{array}]$$

For the INS/DVL/PS integration, hence, all of the states are individually unobservable, except the velocity errors, the altitude error (δh), and the accelerometer bias in the z-axis. Furthermore, the gyro biases in the x ($b_{gx}$) and z ($b_{gz}$) axes are coupled with the latitude error (δL), unlike the result of the INS/GPS/DVL/PS integration shown in Equation (38). In such an integration, $b_{gx}$ and $b_{gz}$ are coupled only with the accelerometer bias in the x-axis ($b_{ax}$).

For the INS/DVL integration, it is possible to show that the rank of $Q_{o}$ is equal to 9, which makes the dimension of the unobservable subspace equal to 10. The kernel of $Q_{o}$, in this case, is as follows:
(43)$$\ker(Q_{0})=[\begin{array}{cccccccccc} 0 & 0 & 0 & 0 & \frac{1}{g} & 0 & 0 & 0 & 0 & 0\\ 0 & 0 & 0 & -\frac{1}{g} & 0 & 0 & 0 & 0 & 0 & 0\\ 0 & \frac{1}{\Omega \cos(L)} & 0 & 0 & -\frac{\tan(L)}{g} & 0 & 0 & 0 & 0 & 0\\ 0 & 0 & 0 & 0 & 0 & 0 & 0 & 0 & 0 & 0\\ 0 & 0 & 0 & 0 & 0 & 0 & 0 & 0 & 0 & 0\\ 0 & 0 & 0 & 0 & 0 & 0 & 0 & 0 & 0 & 0\\ 0 & 0 & -\frac{1}{\Omega \cos(L)} & \frac{1}{g} & 0 & 0 & 0 & 0 & 0 & 0\\ 1 & 0 & 0 & 0 & 0 & 0 & 0 & 0 & 0 & 0\\ 0 & 0 & 0 & 0 & 0 & \frac{R_{0}+h}{2g} & 0 & 0 & 0 & 0\\ 0 & 0 & \tan(L) & 0 & 0 & 0 & 0 & 0 & 0 & 0\\ 0 & 1 & 0 & 0 & 0 & 0 & 0 & 0 & 0 & 0\\ 0 & 0 & 1 & 0 & 0 & 0 & 0 & 0 & 0 & 0\\ 0 & 0 & 0 & 1 & 0 & 0 & 0 & 0 & 0 & 0\\ 0 & 0 & 0 & 0 & 1 & 0 & 0 & 0 & 0 & 0\\ 0 & 0 & 0 & 0 & 0 & 1 & 0 & 0 & 0 & 0\\ 0 & 0 & 0 & 0 & 0 & 0 & 1 & 0 & 0 & 0\\ 0 & 0 & 0 & 0 & 0 & 0 & 0 & 1 & 0 & 0\\ 0 & 0 & 0 & 0 & 0 & 0 & 0 & 0 & 1 & 0\\ 0 & 0 & 0 & 0 & 0 & 0 & 0 & 0 & 0 & 1 \end{array}]$$

The observable states and combinations of states are as follows:(44)$${\bar{x}}_{o}=\left[\begin{array}{c} \phi_{N}-\frac{b_{ay}}{g}\\ \phi_{E}+\frac{b_{ax}}{g}\\ \phi_{D}-\frac{b_{gy}}{\Omega \cos\left(L\right)}+\frac{\tan\left(L\right)b_{ay}}{g}\\ \delta \upsilon_{N}\\ \delta \upsilon_{E}\\ \delta \upsilon_{D}\\ b_{gx}+\Omega sin\left(L\right)\delta L-\frac{\Omega \sin\left(L\right)b_{ax}}{g}\\ b_{gz}+\Omega \;cos\left(L\right)\delta L-\frac{\Omega \cos\left(L\right)b_{ax}}{g}\\ b_{az}-\frac{2g\delta h}{R_{0}+h} \end{array}\right]$$

While the unobservable states and combinations of states are as follows:(45)$${\bar{x}}_{o}=\left[\begin{array}{c} \delta \lambda \\ b_{gy}\\ \frac{\Omega \cos\left(L\right)b_{ax}}{g}-\Omega \;cos\left(L\right)\delta L\\ b_{ax}\\ b_{ay}\\ \frac{2g\delta h}{R_{0}+h}\\ e_{x}\\ e_{y}\\ e_{z}\\ s_{f} \end{array}\right]$$

For the INS/DVL integration, it can be seen that all states are individually unobservable, except for the velocity errors. The gyro biases $b_{gx}$ and $b_{gz}$ are coupled with the latitude error and the accelerometer bias in the x-axis. In addition, $b_{az}$ is coupled with the altitude error, δh, which does not occur in the INS/GPS/DVL/PS and INS/DVL/PS integrations.

The analytical expressions for the observability of the states, obtained for the stationary case, show that the addition of position errors (latitude, longitude, and altitude) in the observation vector not only increases the dimension of the observable subspace, but also decouples $b_{gx}$ and $b_{gz}$ from δL, and $b_{az}$ from δh. In practical terms, decoupling $b_{gx}$, $b_{gz}$, and $b_{az}$ from latitude and altitude errors improves the accuracy of estimation of these variables.

### 6.2. Comparison Results for INS/GPS/DVL/PS Integration

Table 1 shows the results of the system observability analysis performed by the rank computation of $Q_{\mathrm{SOM}}$. For the stationary maneuver, the rank of $Q_{\mathrm{SOM}}$ is 12. This value is in accordance with the rank of $Q_{o}$ obtained in Section 6.1. In addition, it can be seen that the system is observable only for the lawn mower with zigzag and swing maneuver.

The comparative results of the degree of observability for the state variables are illustrated in Figure 1, Figure 2, Figure 3 and Figure 4. Figure 1 shows that the degree of observability of $\phi_{D}$ was higher for the straight-line with the acceleration maneuver (yellow bar), lawn mower maneuvers (green and dark blue bars), and zigzag (middle blue bar). Likewise, $\phi_{N}$ and $\phi_{E}$ obtained a higher degree of observability for lawn mower, straight-line with acceleration, and zigzag.

The lowest degrees of observability of $\phi_{N}$ and $\phi_{E}$ were obtained for stationary (light blue bar) and straight-line with constant speed (gray bar) maneuvers. For $\phi_{D}$, the lowest degrees of observability were obtained for stationary, straight-line with constant speed, and mooring maneuvers (orange bar).

Figure 2 shows that the degrees of observability for $b_{gx}$ and $b_{gy}$ were lower for the stationary and straight-line with constant speed maneuvers, and higher for lawn mower, straight-line with acceleration, and zigzag. Concerning $b_{gz}$, the degree of observability was lower for stationary, mooring, and straight-line with constant speed maneuvers, and was higher for lawn mower and straight-line with acceleration.

Figure 3 shows that the degrees of observability of $b_{ax}$, $b_{ay}$, and $b_{az}$ were higher for lawn mower maneuvers, straight-line with acceleration, zigzag, and mooring. For the stationary maneuver and straight-line with a constant speed, the degrees of observability of the accelerometer biases were lower. In general, it can be seen that maneuvers in which there were no changes of direction or oscillation resulted in lower degrees of observability for the accelerometer biases.

Figure 4 shows that the degrees of observability of $e_{y}$, $e_{z}$, and $s_{f}$ were higher for the straight-line with acceleration maneuver, lawn mower, and zigzag. For the stationary, straight-line with constant speed, and mooring maneuvers, the degrees of observability of $e_{y}$, $e_{z}$, and $s_{f}$ were lower. Regarding $e_{x}$, the degree of observability was higher for the mooring and lawn mower with zigzag and swing maneuvers.

It is interesting to note that the degree of observability of $e_{x}$ was improved for the mooring and lawn mower with zigzag and swing maneuvers, as these scenarios induced velocity components in the y and z axes of the vehicle.

### 6.3. Comparison Results for INS/DVL/PS Integration

Unlike the INS/GPS/DVL/PS integration, Table 2 shows that the INS/DVL/PS integration made the system unobservable for all of the maneuvers evaluated. This result was already expected, as the longitude and latitude errors were unobservable states for INS/DVL/PS integration. Furthermore, for the stationary maneuver, the rank of $Q_{SOM}$ was found to be 10, whose value was in agreement with the rank of $Q_{o}$ obtained in Section 6.1.

The comparative results of the degree of observability for the INS/DVL/PS integration are illustrated in Figure 5, Figure 6, Figure 7 and Figure 8. Figure 5, in particular, shows that the degrees of observability of the attitude errors were higher for lawn mower, zigzag, and straight-line with acceleration maneuvers. The degrees of observability of $\phi_{N}$ and $\phi_{E}$ were lower for stationary and straight-line with constant speed maneuvers. Concerning $\phi_{D}$, the degree of observability was smaller for stationary maneuver, straight-line with constant speed, and mooring.

Figure 6 shows that the lowest degrees of observability for $b_{gx}$ were obtained for the stationary and straight-line with constant speed maneuvers, and the highest degrees for mooring, lawn mower, and zigzag maneuvers. Regarding $b_{gy}$, the lowest degrees of observability were obtained for the stationary, mooring, and straight-line maneuvers, and the highest degrees were for lawn mower and zigzag. For $b_{gz}$, the lowest degrees of observability were obtained for the stationary, mooring, and straight-line with constant speed maneuvers, and the highest were for lawn mower, zigzag, and straight-line with acceleration maneuvers.

Concerning the accelerometer biases, Figure 7 shows that the degree of observability of $b_{ax}$ was higher for lawn mower maneuvers, straight line with acceleration, and zigzag, and lower for stationary, mooring, and straight line with constant speed maneuvers. For $b_{ay}$ and $b_{az}$, the degree of observability was higher for maneuvers with acceleration or oscillation. It is worthy of note that, although the degrees of observability of the accelerometer biases were increased with lawn mower and zigzag maneuvers, the differences with respect to the remaining maneuvers were not significant.

Lastly, and as also observed for INS/GPS/DVL/PS integration, Figure 8 shows that the degrees of observability of $e_{y}$, $e_{z}$, and $s_{f}$ were higher for the lawn mower, zigzag, and straight-line with acceleration maneuvers. The degrees of observability were lower, in turn, for the stationary, straight-line with constant speed, and mooring scenarios. Regarding $e_{x}$, the degree of observability was higher for the mooring and lawn mower with zigzag and swing maneuvers.

### 6.4. Comparison Results for INS/DVL Integration

Table 3 shows that the system was unobservable for all of the maneuvers evaluated using the simplified INS/DVL integration scheme. This result was again already expected as the position errors were unobservable states for the INS/DVL integration. In addition, for stationary maneuver, the rank of $Q_{\mathrm{SOM}}$ was 9, whose value was equal to the rank of $Q_{o}$ obtained in Section 6.1.

Figure 9, Figure 10, Figure 11 and Figure 12 show the comparative results of the degree of observability for the INS/DVL integration. As illustrated in Figure 9, the highest degrees of observability for $\phi_{N}$ and $\phi_{E}$ were obtained for the zigzag maneuver. The lowest degrees of observability for $\phi_{N}$ and $\phi_{E}$, in turn, were obtained for the stationary maneuver. Regarding $\phi_{D}$, the highest degrees of observability were obtained for the lawn mower and zigzag maneuvers. On the contrary, the lowest degrees were obtained for the stationary, mooring, and straight-line with constant speed scenarios.

Concerning the gyro biases, the degrees of observability were lower for the stationary maneuver, and higher for zigzag, as illustrated in Figure 10. For the accelerometer biases, Figure 11 shows that the degrees of observability were higher for zigzag and mooring ($b_{az}$), and lower for the stationary maneuver.

Concerning $e_{y}$, $e_{z}$, and $s_{f}$, Figure 12 shows that the degrees of observability were higher for the maneuvers of lawn mower, zigzag, and straight-line with acceleration. The degrees of observability were lower, however, for the stationary maneuver, straight-line with constant speed, and mooring. Regarding $e_{x}$, finally, its degree of observability was found to be higher for the mooring and lawn mower with zigzag and swing maneuvers.

## 7. Discussion

The observability decomposition performed for the stationary case showed that the addition of the position errors in the observation vector decoupled the biases $b_{gx}$, $b_{gz}$, and $b_{az}$ from the latitude ($b_{gx}$ and $b_{gz}$) and altitude errors ($b_{az}$). Furthermore, maneuvers including changes of direction and speed excitation in the three axes increased the dimension of the observable subspace. For the INS/GPS/DVL/PS integration, in particular, the system became fully observable for these types of maneuvers.

In addition, the simulation results showed that the alignment errors obtained higher degrees of observability for the lawn mower, zigzag (especially for INS/DVL), and straight-line with acceleration maneuvers, while the lower degrees of observability for the stationary and straight-line at constant speed maneuvers. In particular, $\phi_{D}$ was shown to possess the lowest degree of observability among the attitude errors.

For the three integration schemes evaluated, the degree of observability of $\phi_{D}$ was increased by accelerating the AUV in the local navigation frame, especially in the horizontal plane (which corresponds to the straight-line with acceleration, zigzag, and lawn mower maneuvers). For the lawn mower with zigzag and swing maneuver, the change of orientation in the horizontal plane additionally provoked velocity variations in the NED frame, which produced acceleration components in the north and east directions. The influence of such accelerations on the degree of observability of $\phi_{D}$ has also been observed by other authors [12,14,15,27].

Regarding the gyro biases, the simulation results showed that the degree of observability of $b_{gz}$ was increased with vertical oscillation or acceleration in the x-axis. Particularly for INS/DVL integration, the degree of observability of $b_{gz}$ was higher for the zigzag maneuver. Furthermore, the zigzag and lawn mower maneuvers also improved the degrees of observability of the accelerometer biases. As demonstrated in [18], the estimations of the accelerometer biases were improved with angular movements.

Finally, the degrees of observability of the IMU/DVL misalignment in the y and z axes, as well as, the DVL scale factor error, were significantly increased by the occurrence of translational motion. Specifically, for $e_{x}$, the degree of observability was improved with translational movements in the y and z axes of the AUV. This condition was also evidenced in [19] for a simplified model, in which the orientation and the velocity of the vehicle were perfectly known. In [18], it was demonstrated that the AUV displacement in the z-axis with constant attitude enabled the estimation of $e_{x}$ in single-thruster AUVs. It is noteworthy that $e_{x}$, $e_{y}$, $e_{z}$ and $s_{f}$ were individually unobservable for the stationary case, regardless of the three integrations schemes evaluated, which means that the calibration of the DVL with stationary AUV is not possible to be performed.

Despite being relevant, the degree of observability analysis did not allow us to visualize the couplings between the state variables. This has only been possible for the stationary case, via observability decomposition, which evidenced the existence of the couplings $\left\{\phi_{N},b_{ay}\right\}$ and $\left\{\phi_{E},b_{ax}\right\}$, as well as the couplings between $\phi_{D}$, $b_{gy}$, and $b_{ay}$. These results are in agreement with the traditional literature [21,22,23,28].

Additionally, the observability decomposition for the stationary case showed that $b_{ax}$, $b_{ay}$, and $b_{gy}$ are variable states of the unobservable subspace. Regarding $b_{gx}$ and $b_{gz}$, they can be considered individually observable, as their couplings with $b_{ax}$, that is, $\left\{b_{gx},-\frac{\Omega \sin\left(L\right)b_{ax}}{g}\right\}$ and $\left\{b_{gz},-\frac{\Omega \cos\left(L\right)b_{ax}}{g}\right\}$, are negligible, given that the magnitudes of the weighting coefficients of $b_{ax}$ are less than $10^{-5}$ rad.s/m.

It is also noteworthy that the degree of observability obtained for $b_{gy}$ was close to $b_{gx}$ and significantly higher than $b_{gz}$. Such results are in agreement with previous works [12,14,15], but raises the question that the degree of observability may not be directly related to the estimation accuracy of the states, given that $b_{gx}$ and $b_{gz}$ are recognizably known to be better estimated than $b_{gy}$ in the AUV initial alignment and calibration problem [23].

Therefore, the simulation results presented in this paper suggest that the exclusive analysis of the degree of observability may not be a suitable metrics for the “estimability” of the states. As defined by Baram and Kailath [29], the estimability of a system is the ability to reduce the mean squared error in the estimation of the states. As suggested by the authors, the estimability analysis can be seen as a complementary tool to the observability analysis. In [9], for instance, observability and estimability analyses were carried out separately for the space situational awareness (SSA) problem. The observability analysis, in terms of the observability matrix rank, aimed to inform whether the measurements were sufficient for the estimation of the states. Conversely, the estimability analysis aimed to evaluate the performance of the estimation.

## 8. Conclusions

In this work, the effects of different maneuvers and integration schemes on the AUV initial alignment and calibration were analyzed in terms of system’s observability and state variables’ degree of observability. The simulation results showed that the system became fully observable for INS/GPS/DVL/PS integration when lawn mower with zigzag and swing maneuvers were carried out.

For the three investigated integrations, the observability degrees of the IMU/DVL misalignment and the DVL scale factor error were increased with translational motion. In addition, the degrees of observability of the inertial sensor biases and the vertical alignment error were increased with maneuvers encompassing changes of direction. The observability degrees of the vertical alignment error and the gyro bias in the z-axis, lastly, were increased by accelerating the AUV along its longitudinal axis.

Finally, the simulation results presented in this article suggest that the exclusive analysis of the degree of observability may not be an adequate metric for the “estimability” of the states. In this sense, a thorough investigation on the physical meaning of the degree of observability and its relation with the estimability of states, specifically in the scope of the AUV initial alignment and calibration problem, will be a topic for future investigation by the authors.

## Figures and Tables

**Figure 1 sensors-22-03287-f001:**
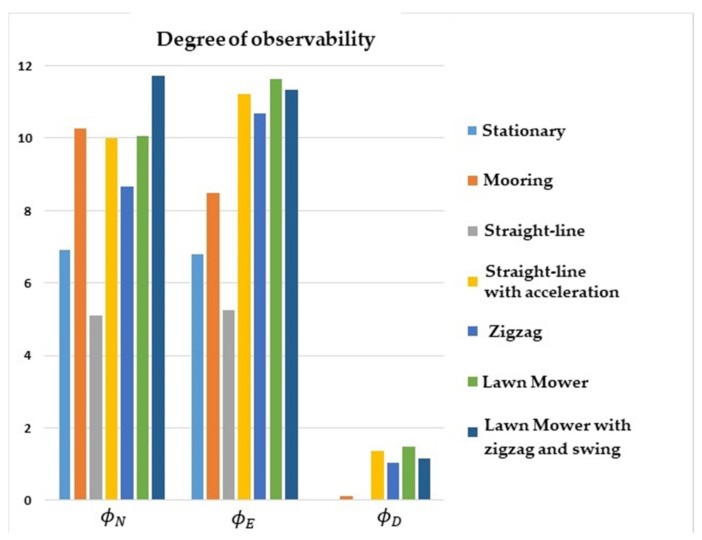
Degrees of observability of the attitude errors. INS/GPS/DVL/PS integration.

**Figure 2 sensors-22-03287-f002:**
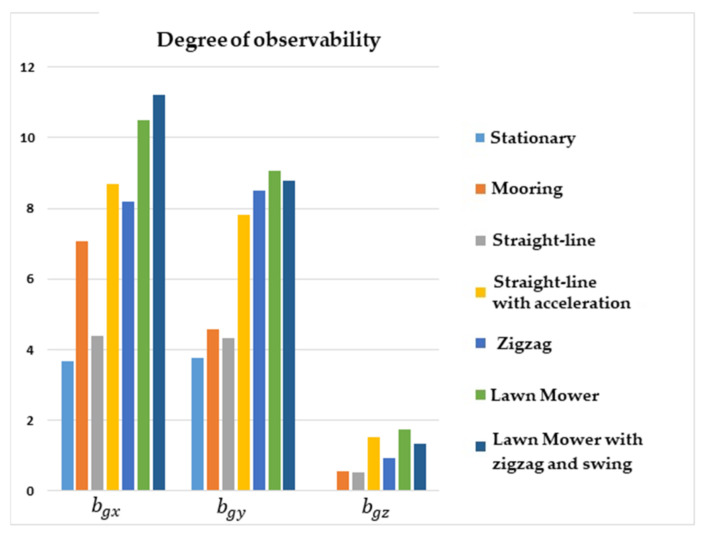
Degrees of observability of the gyro biases. INS/GPS/DVL/PS integration.

**Figure 3 sensors-22-03287-f003:**
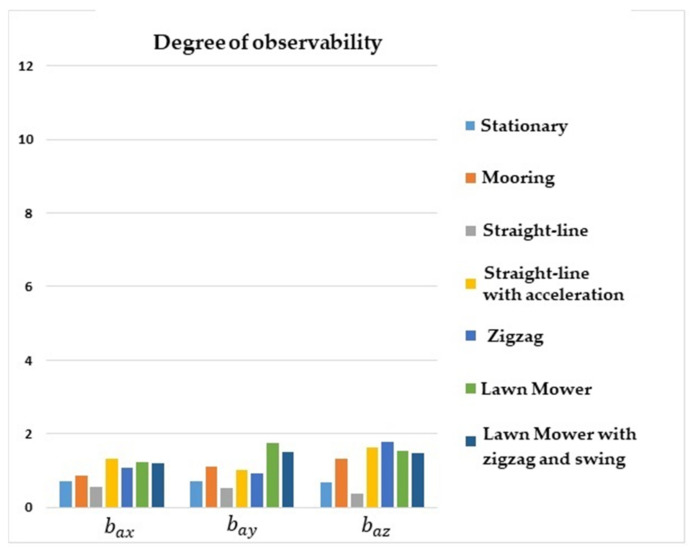
Degrees of observability of the accelerometer biases. INS/GPS/DVL/PS integration.

**Figure 4 sensors-22-03287-f004:**
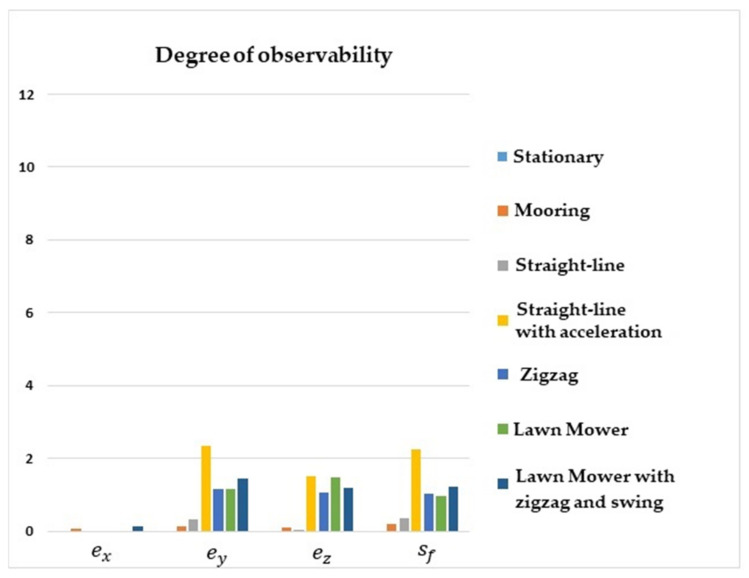
Degrees of observability of the IMU/DVL misalignment and the DVL scale factor error. INS/GPS/DVL/PS integration.

**Figure 5 sensors-22-03287-f005:**
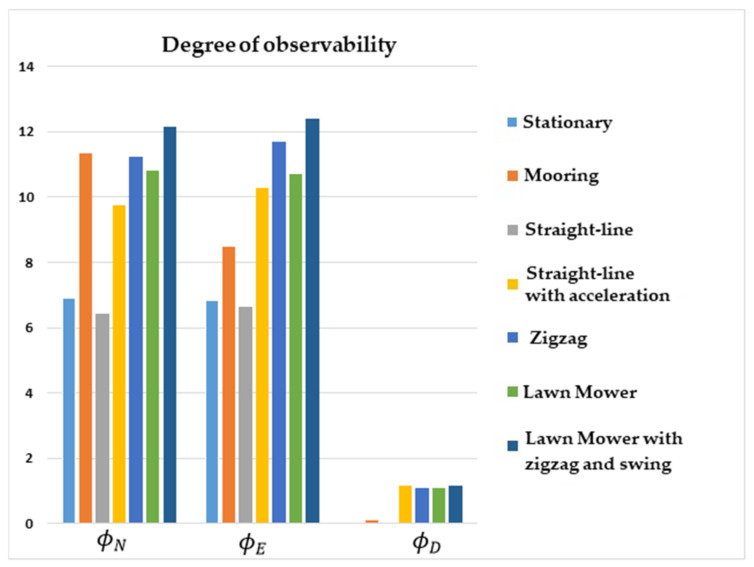
Degrees of observability of the attitude errors. INS/DVL/PS integration.

**Figure 6 sensors-22-03287-f006:**
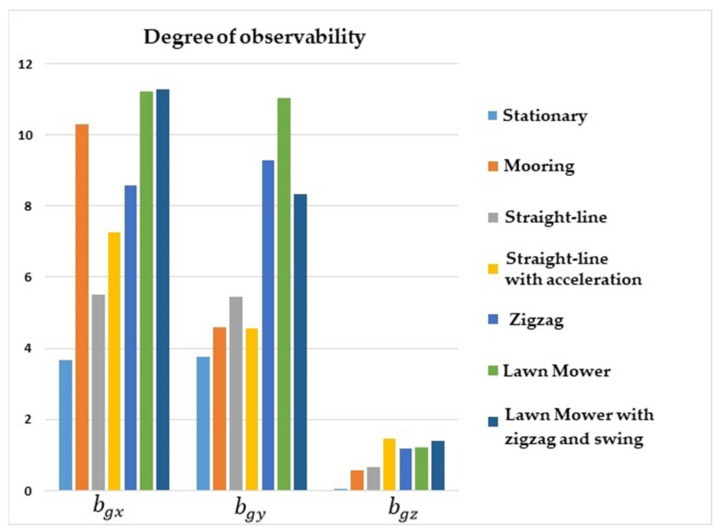
Degrees of observability of the gyro biases. INS/DVL/PS integration.

**Figure 7 sensors-22-03287-f007:**
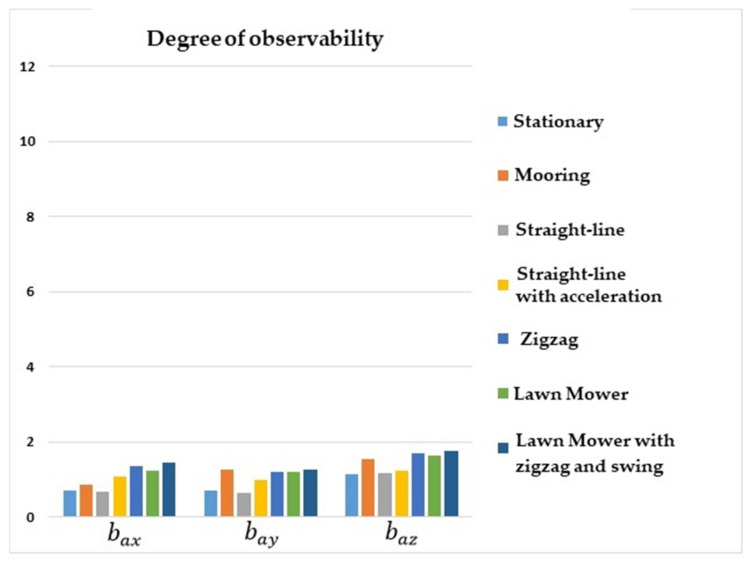
Degrees of observability of the accelerometer biases. INS/DVL/PS integration.

**Figure 8 sensors-22-03287-f008:**
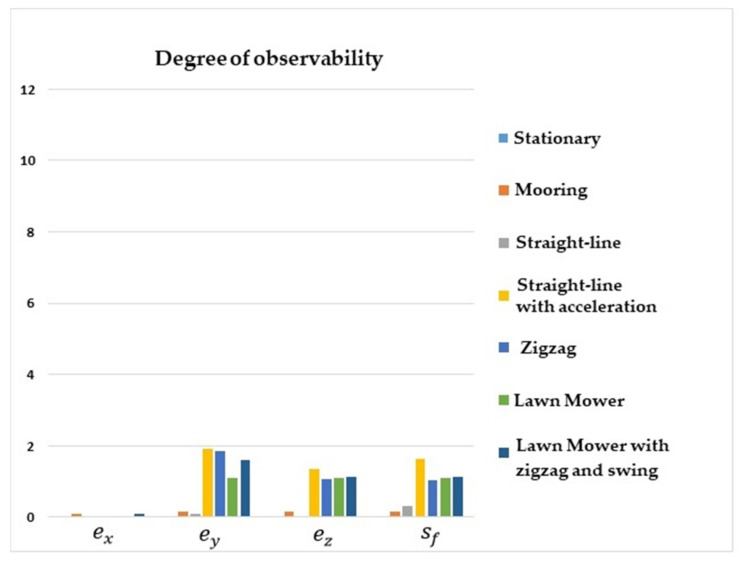
Degrees of observability of the IMU/DVL misalignment and the DVL scale factor error. INS/DVL/PS integration.

**Figure 9 sensors-22-03287-f009:**
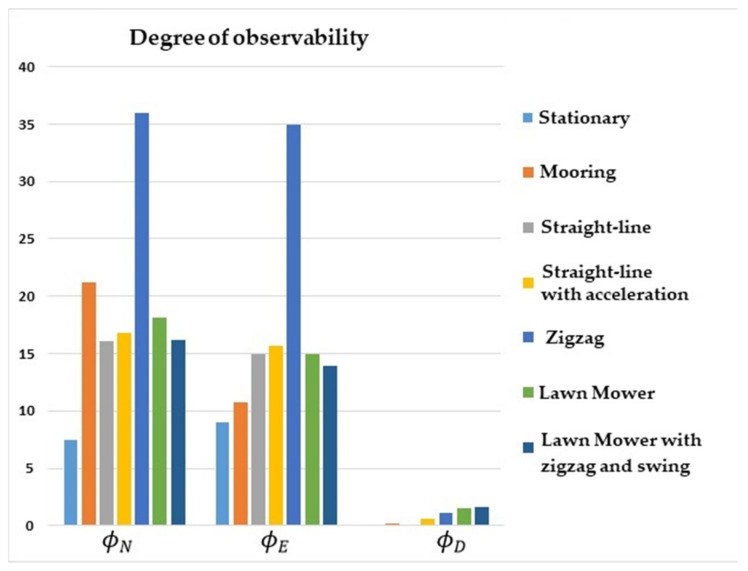
Degrees of observability of the attitude errors. INS/DVL integration.

**Figure 10 sensors-22-03287-f010:**
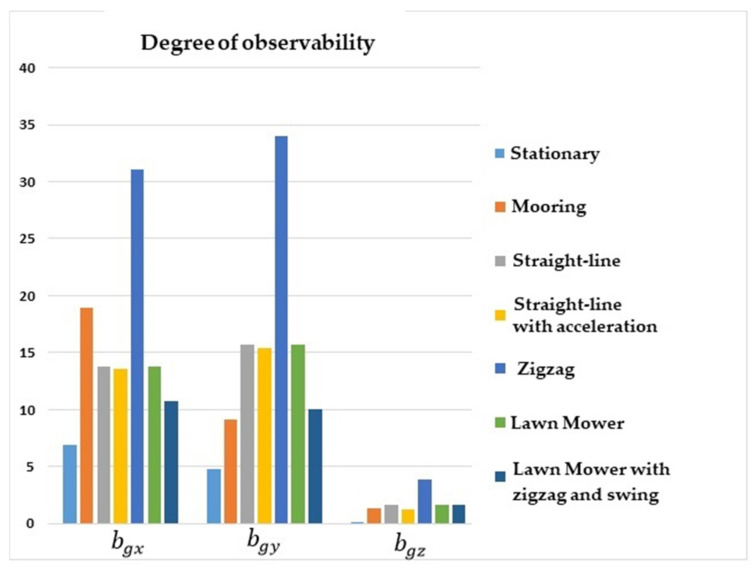
Degrees of observability of the gyro biases. INS/DVL integration.

**Figure 11 sensors-22-03287-f011:**
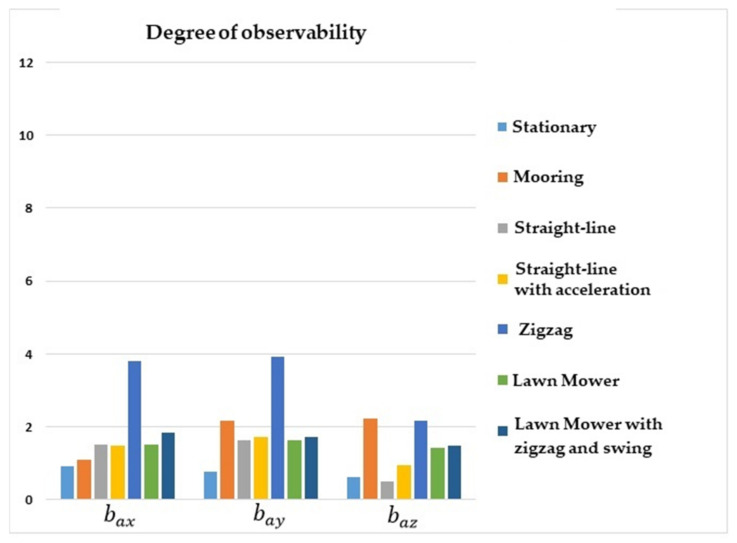
Degrees of observability of the accelerometer biases. INS/DVL integration.

**Figure 12 sensors-22-03287-f012:**
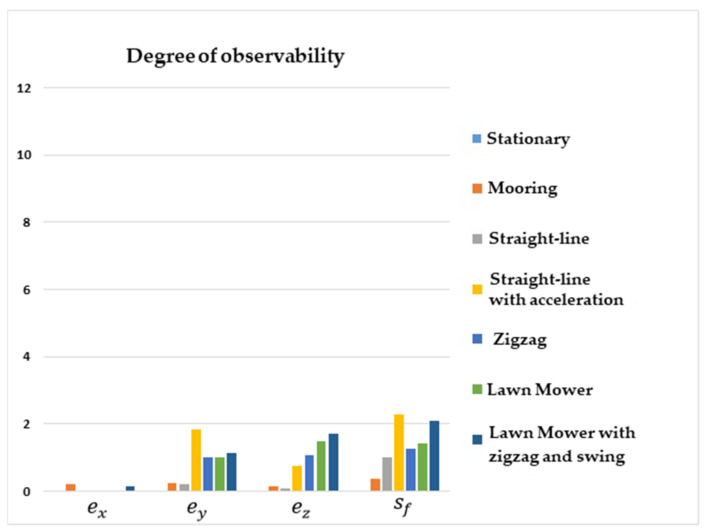
Degrees of observability of the IMU/DVL misalignment and the DVL scale factor error. INS/DVL integration.

**Table 1 sensors-22-03287-t001:** Observability and final rank of $Q_{\mathrm{SOM}}$ (j=100). INS/GPS/DVL/PS integration.

Maneuver	$Q_{\mathrm{SOM}}\;\mathrm{Rank}$	System Observability
Stationary	12	Unobservable
At Mooring	18	Unobservable
Straight-line	18	Unobservable
Straight-line with acceleration	18	Unobservable
Zigzag	18	Unobservable
Lawn Mower	18	Unobservable
Lawn Mower with zigzag and swing	19	Observable

**Table 2 sensors-22-03287-t002:** Observability and final rank of $Q_{SOM}$ (j=100). INS/DVL/PS integration.

Maneuver	$Q_{\mathrm{SOM}}\;\mathrm{Rank}$	System Observability
Stationary	10	Unobservable
At Mooring	17	Unobservable
Straight-line	16	Unobservable
Straight-line with acceleration	17	Unobservable
Zigzag	17	Unobservable
Lawn Mower	17	Unobservable
Lawn Mower with zigzag and swing	18	Unobservable

**Table 3 sensors-22-03287-t003:** Observability and final rank of $Q_{\mathrm{SOM}}$ (j=100). INS/DVL integration.

Maneuver	$Q_{\mathrm{SOM}}\;\mathrm{Rank}$	System Observability
Stationary	9	Unobservable
At Mooring	17	Unobservable
Straight-line	15	Unobservable
Straight-line with acceleration	16	Unobservable
Zigzag	16	Unobservable
Lawn Mower	16	Unobservable
Lawn Mower with zigzag and swing	17	Unobservable

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
