# Peer review of "Influence of Integration Schemes and Maneuvers on the Initial Alignment and Calibration of AUVs: Observability and Degree of Observability Analyses"

_sensors, 2022, doi:10.3390/s22093287_

Round 1

Reviewer 1 Report

 the authors propose a thorough observability analysis applied to the latter problem. The observability analysis is carried out considering three types of sensor fusion integration and a set of maneuvers, and the results are validated through numerical simulations. the work is interested however, there are few comments to improve the paper 

1-What about the difference between underwater vehicles, ground and aerial vehicles, authors should discuss this in the introduction and related work. what about the applications such as Predictive estimation of the optimal signal strength from unmanned aerial vehicle over internet of things using ANN,Performance optimization of tethered balloon technology for public safety and emergency communications,Disaster coverage predication for the emerging tethered balloon technology: capability for preparedness, detection, mitigation, and response, Green IoT for eco-friendly and sustainable smart cities: future directions and opportunities,Green internet of things using UAVs in B5G networks: a review of applications and strategies

2-figures and equations organization should be done 

3- authors need to improve the paper by discussing the most current work that are missing such as Convergence of machine learning and robotics communication in collaborative assembly: mobility, connectivity and future perspectives,Survey on artificial intelligence based techniques for emerging robotic communication. 

4-figures are poor resolution so please try to make it in high resolution

5-separate conclusion should be one section and discussion and results in one section

3-conclusion should include your main contribution findings and few lines about future work of current work in one paragraph 

Reviewer 2 Report

Please to see the attached file. Thank you.
